# Prolonged Mechanical Ventilation: Outcomes and Management

**DOI:** 10.3390/jcm11092451

**Published:** 2022-04-27

**Authors:** Hung-Yu Huang, Chih-Yu Huang, Li-Fu Li

**Affiliations:** 1Division of Pulmonary and Critical Care Medicine, Department of Internal Medicine, Chang Gung Memorial Hospital, Taoyuan 333, Taiwan; compaction71@gmail.com; 2Department of Internal Medicine, College of Medicine, Chang Gung University, Taoyuan 333, Taiwan; huv71@yahoo.com.tw; 3Department of Thoracic Medicine, New Taipei City Municipal Tucheng Hospital, Chang Gung Medical Foundation, New Taipei City 236, Taiwan; 4Division of Pulmonary and Critical Care Medicine, Department of Internal Medicine, Chang Gung Memorial Hospital, 222 Mai-Chin Road, Keelung 204, Taiwan

**Keywords:** prolonged mechanical ventilation, reactive oxygen species, respiratory drive, ventilator-induced diaphragm dysfunction

## Abstract

The number of patients requiring prolonged mechanical ventilation (PMV) is increasing worldwide, placing a burden on healthcare systems. Therefore, investigating the pathophysiology, risk factors, and treatment for PMV is crucial. Various underlying comorbidities have been associated with PMV. The pathophysiology of PMV includes the presence of an abnormal respiratory drive or ventilator-induced diaphragm dysfunction. Numerous studies have demonstrated that ventilator-induced diaphragm dysfunction is related to increases in in-hospital deaths, nosocomial pneumonia, oxidative stress, lung tissue hypoxia, ventilator dependence, and costs. Thus far, the pathophysiologic evidence for PMV has been derived from clinical human studies and experimental studies in animals. Moreover, recent studies have demonstrated the outcome benefits of pharmacological agents and rehabilitative programs for patients requiring PMV. However, methodological limitations affected these studies. Controlled prospective studies with an adequate number of participants are necessary to provide evidence of the mechanism, prognosis, and treatment of PMV. The great epidemiologic impact of PMV and the potential development of treatment make this a key research field.

## 1. Definition and Clinical Prevalence

A percentage of critically ill patients experience chronic respiratory failure and require prolonged mechanical ventilation (PMV) [1]. PMV is defined as successful extubation after more than three spontaneous breathing trials or taking more than 14 days [1], although some studies have used different definitions, especially studies in the United Kingdom and Europe [2]. Approximately 5–13% of patients with acute respiratory failure require PMV, and a trend of increasing PMV exists worldwide [3]. In the United States, the number of patients requiring PMV was estimated to be approximately 625,000 in 2020 [4]. In Canada, approximately 11% of ventilator-capable beds in intensive care units (ICUs) are for patients who require PMV [5], and in Taiwan, the incidence rate of PMV has increased, from 50,481 patients receiving PMV between 1997 and 2007 to 94,324 patients between 2015 and 2019 in total [6,7].

## 2. Pathophysiology of PMV

Patients receiving PMV may experience complications, including limb muscle atrophy, impaired functional status, and diaphragm dysfunction [8]. Moreover, the basal respiratory drive levels may be altered in patients with PMV, depending on the pulmonary disease present and the cause of respiratory failure [9,10,11]. Brain stem lesions may impair the central and low respiratory drives, causing abnormal respiratory function, hypoventilation, or respiratory acidosis, any of which may lead to ventilator dependence [11]. 

## 3. Respiratory Drive

The brain stem responds to levels of O_2_, CO_2_, and pH in the blood and cerebrospinal fluid through chemoreceptive neurons and initiates automatic respiration [12]. Airway occlusion pressure (P0.1) or hypercapnic ventilatory response can be used to measure respiratory drive [9,11,13]. P0.1 is defined as the negative airway pressure generated during the first 100 ms of an occluded inspiration measured by a ventilator [13]. The hypercapnic challenge test is performed with the modified Read rebreathing method, and the hypercapnic challenge with CO_2_ induces an increase in the ventilatory drive and the level of P0.1 recorded by bedside capnograph and pneumotachograph [13]. The hypercapnic ventilatory response is used as a predictor of weaning outcomes in patients with short- or long-term use of MV; old age, comorbidities, and muscle weakness, which may lead to a low hypercapnic ventilatory response [11,14,15,16]. In patients receiving PMV with brainstem lesions (brain stem infarction, brain tumor, traumatic intracerebral hemorrhage), high P0.1 responses to a hypercapnic challenge are associated with higher rates of successful weaning from MV, and an increase in P0.1 of more than 6 cmH_2_O after the hypercapnic challenge is a predictor of weaning success [11]. 

Mechanical power is the energy transferred to the respiratory system per unit of time, which is the product of respiratory rate, tidal volume, and the airway pressure during pressure-controlled ventilation [17,18]. Higher mechanical power is associated with ventilator-induced lung injury and worse clinical outcomes [19]. Mechanical power is a novel concept to guide mechanical ventilator adjustments for acute respiratory distress syndrome [20]. For patients with PMV, mechanical power may be a tool used as respiratory drive to assess ventilator weaning and needs further investigation.

## 4. Ventilator-Induced Diaphragm Dysfunction

The diaphragm is the primary skeletal muscle responsible for effective lung expansion [21]. The diaphragm is a thin, dome-shaped muscular and central membranous tendon structure that separates the thoracic and abdominal cavities. Under normal conditions, contraction of the diaphragm increases the internal height of the chest cavity and causes the inspiration of air. Relaxation of the diaphragm and the thoracic cage leads to expiration. The diaphragm strength is measured by the transdiaphragmatic pressure (Pdi) derived from the difference between the pressure in the stomach (gastric pressure, Pga) and the esophageal pressure (Pes, as an indicator of pleural pressure): Pdi = Pga − Pes. The gold standard in patients who cannot cooperate in the test is to calculate the Pdi via twitch magnetic stimulation of the phrenic nerves (Pdi,tw). Maximal inspiratory pressure (MIP) is recorded following a maximum inspiratory effort against an occluded airway [22,23,24]. PMV may induce the rapid debilitation of diaphragmatic strength and endurance and lead to a reliance on ICU ventilators or long-term facilities; Petrof and Vassilopoulos refer to this as ventilator-induced diaphragm dysfunction (VIDD) [25,26,27,28]. Demoule et al. revealed that approximately 80% of ICU patients receiving MV exhibit an altered pattern of ICU-acquired diaphragm weakness after the first use of MV [28]. Numerous studies have demonstrated that VIDD is associated with increases in in-hospital deaths, nosocomial pneumonia, oxidative stress, lung tissue hypoxia, ventilator dependence, and costs [22,23,28], suggesting that diaphragmatic weakness with a reduced capacity of the diaphragm to produce inspiratory pressure may be considered as an example of unacknowledged organ failure in patients with a critical illness. Respiratory muscle abnormalities in critically ill patients may arise from tissue hypoxia, abnormally high diaphragm activity, neuromyopathies, hypercapnic respiratory failure, sepsis, medications (sedatives, steroids, and neuromuscular blocking reagents), MV duration, and malnutrition [24,29]. Sepsis is linked to diaphragm dysfunction in more than half of the ICU patients with PMV [24,30]. Tissue hypoxia is frequently associated with inflammation in patients with sepsis, which is induced by several mechanisms, including microvascular hypoperfusion, microthrombi formation, and regional arteriolar vasoconstriction related to hypercoagulability and leucosequestration [24,30]. Previous animal and brain-dead human studies have revealed that the onset of diaphragm injury was rapid, occurring within 6 to 18 h after MV and the magnitude of impairment of diaphragmatic contraction increased with the time spent on the ventilator [23,24,25,26]. Because of the high medical care expenses and poor prognosis associated with delayed extubation and a prolonged hospital stay of PMV patients, it is, therefore, an important objective to assist ventilator-dependent patients in early weaning from ventilators. Although the specific etiology of PMV is unclear, long-term use of MV may result in the rapid development of VIDD because of concurrent critical ill neuromyopathy, severe infection, excessive or low-pressure support, and patient-ventilator asynchrony [23,24,25,26]. The underlying pathophysiological mechanisms posited for the decrease in diaphragm muscle contractility and endurance are thought to result from elevated oxidative stress, muscle proteolysis (arising from ubiquitin–proteasome system activation, calpain, caspase-3, and the autophagy–lysosomal pathway), and mitochondrial injuries within the diaphragm muscle fibers [24,26]. Figure 1 illustrates the signaling pathway implicated in VIDD development.

Reactive oxygen species (ROS), which are produced in mitochondria, sarcoplasmic reticula, sarcolemma, and transverse tubes, are the major oxidants and upstream regulators of proteolysis and mitochondrial dysfunction in the diaphragm [31,32]. Caspase-3 is a cysteine protease-evoking skeletal muscle proteolysis, which is initiated by class O of forkhead box 1 (FoxO1) and the oxidative load in VIDD [33,34,35]. When proteolysis is initiated, caspase-3 can disassemble actin and myosin complexes from the myofibrillar lattice and trigger the release of dissociated myofibrillar proteins which are susceptible to degeneration through the ubiquitin–proteasome system. Calpains are reported to be responsible for the occurrence of VIDD, and their primary function is to support myofibrillar protein turnover by releasing sarcomeric proteins for degradation by caspase-3 [29,36].

An MV-induced oxidative load may impair diaphragm contractility, which is vital for increased proteolytic pathway expression [37,38,39]. The principal proteases in skeletal muscles comprise the ubiquitin–proteasome system and lysosomal enzymes [24,40]. The upregulation of the diaphragm muscle-specific ubiquitin E3 ligases, atrogin-1 and RING-finger proteins-1, is crucial for the proteolysis of monomeric myofibrillar proteins in the diaphragms of patients and animals receiving MV [39,40,41]. Mitochondria are major resources of diaphragmatic ROS and serve as an essential upstream modulator that mediates molecular pathways, promoting diaphragm muscle atrophy during endotoxemia or MV [42,43]. Reduced mitochondrial biogenesis and cytochrome-c oxidase enzyme activity, but increased mitochondrial DNA activity and lipid accumulation were identified in the diaphragm of brain-dead patients receiving MV [44]. The respiratory chain complexes II, III, and IV, and the ratio of state 3 to 4 (respiratory control ratio) are reduced in mitochondria isolated from rodent diaphragms [43,45]. Elevated levels of autophagosomes also occur in MV-mediated diaphragmatic inactivity, as reflected in an increase in the autophagic marker microtubule-associated protein light chain 3 [29,46,47].

Nuclear factor-κB (NF-κB) activation has been implicated in diaphragm injury and atrophy through measurements of the maximal twitch airway pressure after magnetic stimulation of the phrenic nerve of patients receiving MV [29,36,48]. An oxidative load triggered by mediators produced from inflammatory cells may react with redox-sensitive NF-κB, leading to the progression of coagulation and inflammation, which are related to the pathogenesis of VIDD or sepsis [49,50]. Previous studies also demonstrated a severe O2 delivery-to-demand mismatch within the diaphragm of mice after PMV, suggesting the potential for diaphragm tissue hypoxia.18,46 Furthermore, Smuder et al. reported that administration of a specific NF-kB inhibitor SN-50 or antioxidant attenuated the MV-induced oxidative stress in the diaphragm and its downstream target NF-κB [32,40]. Oliveira et al. revealed that the diminished expression of detoxifying ROS enzymes (e.g., superoxide dismutase 2) and systemic inflammatory reactions through NF-κB signaling led to a significant decrease in peroxisome proliferator-activated receptor coactivator-1α (a transcriptional coactivator implicated in the regulation of mitochondrial oxidative capacity) expression in the septic diaphragm [38,42]. NF-κB has also been revealed to regulate autophagy in murine skeletal muscles during sepsis. The Smad3, Stat3, FoxO, interleukin (IL)-6, and transforming growth factor-β1 signaling network has been demonstrated to modulate VIDD through periodic phrenic nerve stimulation of diaphragms subjected to MV [41,51]. The downregulation of the phosphoinositide 3-OH kinase, serine, threonine kinase, protein kinase B, and mammalian target of the rapamycin signaling pathway is primarily involved in regulating the processes linked with protein synthesis and protein degradation in skeletal muscle fibers during prolonged periods of inactivity [52]. Several studies have demonstrated that MV augments the expression of hypoxia-inducible factor-1α (HIF-1α) in animal diaphragm tissues [53,54,55]. HIF-1α signaling is closely associated with the vascular response to sepsis [56,57]. HIF-1α is also a principal regulator that controls the transcription of genes that encode proteins that affect metabolic adaptation (glycolytic enzymes and glucose transporters), oxygen delivery (erythropoietin), vascular endothelial growth factor (VEGF)-related angiogenesis, and cell existence (insulin-like growth factor 1). Cyclic stretch, as a result of MV, was revealed to inhibit succinate dehydrogenase activity and escalate succinate production, leading to the suppression of prolyl hydroxylase, a reduction in polyubiquitination and proteasomal degradation, and acceleration of HIF-1α stabilization in vitro and in vivo [58].

In addition to the maximal inspiratory pressure and magnetic stimulation of the phrenic nerves, ultrasonography, a noninvasive tool, is used to evaluate the diaphragm movements of patients requiring PMV [23,26]. Several studies on ICU patients have demonstrated that the diaphragm thickness—analyzed through ultrasound—is correlated with lower respiratory system compliance and the probability of successful weaning, higher rapid shallow breathing index (the ratio of tidal volume in liters to respiratory rate in breaths/minute), an extended ICU stay, and a high risk of ventilator-associated complications [22,45,59,60,61]. Furthermore, the introduction of neurally adjusted ventilatory assist (NAVA), a ventilator mode that synchronizes MV to diaphragm electrical signal using an esophageal catheter with multiple electrodes, has facilitated the application of electromyography in the continuous monitoring of diaphragm activity [24,62].

The current approach to reducing VIDD is to avoid controlled MV and the use of unnecessary neuromuscular blocking agents. Notably, low-molecular-weight heparin (LMWH) can reduce lipopolysaccharide-induced acute lung injury and systemic inflammation in animal models of endotoxemia [56,57]. A randomized clinical trial that included 20 patients demonstrated that the administration of subcutaneous enoxaparin improved serial gas exchange and reduced D-dimer levels, and resulted in a higher percentage of successful weaning in patients with severe COVID-19 requiring PMV [63]. In addition, LMWH was determined to reduce pulmonary inflammation through the suppression of VEGF and HIF-1α expression in an animal model of peritoneal fibrosis, indicating that LMWH provides benefits against hypoxia and inflammation [64]. Challenges for the future include establishing PMV strategies or methods that attenuate the likelihood of VIDD and determining the efficacy of pharmacological interventions, such as antioxidant therapy or inhibitors of muscle proteolysis pathways, in maintaining diaphragmatic function in patients requiring MV [61].

## 5. Clinical Impact and Multiple Comorbidities

### 5.1. Outcome and Prognostic Factors

The clinical outcomes for patients requiring PMV globally have been summarized through a systemic review and meta-analysis [1]: 50% of patients (95% confidence interval [CI]: 47–53) were successfully weaned from MV, in-hospital mortality was 29% (95% CI: 26–32), and only 19% (95% CI: 16–24) of patients were discharged home [1]. Recognized factors of weaning failure are underlying respiratory diseases, previous ICU admissions, a high acute physiology and chronic health evaluation (APACHE II) score, and pneumonia [65,66,67,68]. Other factors related to unsuccessful weaning are elevated blood urea nitrogen levels, low Glasgow coma scale (GCS) scores, low serum albumin, and low maximal inspiratory pressure levels [69].

The prognosis for patients requiring PMV is poor. Among 29 studies of PMV, the pooled mortality rate was 62% at year 1 [1]. In a cohort study in the United States, 53.7% of patients requiring PMV were successfully weaned from ventilation at discharge, and 66.9% of these patients were still alive at year 1 [70]. However, the survival rate of patients with ventilator dependence was only 16.4% at post-discharge year 1 [70].

### 5.2. Comorbidities

Comorbidities exacerbate diseases and have a severe impact on the outcomes of patients requiring PMV. Chronic obstructive pulmonary disease (COPD), cardiac disease, cerebral vascular or neuromuscular disease, end-stage renal disease, and malignancy are common comorbidities in patients requiring PMV [69,71,72,73,74]. Risk factors of PMV are summarized in Table 1.

Patients with PMV and kidney injury requiring renal replacement therapy (RRT) have a lower weaning rate and poor prognosis [71]. PMV patients with RRT have approximately a 2-fold risk of mortality (54.1% vs. 28.2%) compared with those without RRT [71]. Patients with kidney injury requiring PMV and RRT have lower baseline conditions, such as a high APACHE II score and low serum albumin, and kidney injury is associated with the deterioration of respiratory failure [71,75,76,77].

The prognosis for patients with cancer requiring PMV is generally poorer than that for patients with other comorbidities [73]. Patients with cancer requiring PMV generally survive for less than 2 months, and the overall 1-year survival rate is 14.3% [73]. Patients requiring PMV with cancer of the head, neck, esophagus, skin, or nervous system have a more positive prognosis than those with lung cancer or neurogenic tumor [73].

Cardiac dysfunction is a common cause of unsuccessful weaning and PMV [78]. Patients with coronary artery disease and heart failure are also at risk of weaning-induced cardiac dysfunction [79]. During PMV, patients with cardiovascular diseases have a lower successful weaning rate than other etiologies [69]. During the weaning process, with decreased pressure support, respiratory distress results from the increase in the work of breathing, sympathetic tone, or myocardial oxygen demand [80]. During spontaneous breathing, ventilator support is shifted from positive to negative pressure ventilation, and this decrease in intrathoracic pressure may increase the preload or afterload of the left ventricle or decrease left ventricle compliance [81].

Patients with acute stroke or brain trauma may have impaired consciousness, hypoventilation, or aspiration [82,83,84,85], and 18–32% of patients with acute stroke or brain trauma require intubation and MV for respiratory failure, with approximately 10% of stroke survivors requiring PMV [84,86]. The hospital mortality rate for patients who require MV for acute stroke or brain trauma ranges from 16% to 33% [84,87], and 66–76% of patients with acute stroke or brain trauma have severe coma (GCS score < 8) [84]. During ventilator use, some patients with acute stroke or brain trauma may have improved consciousness during hospitalization [74], however reduced GCS scores or persistent consciousness impairment in the weaning center are risk factors for weaning failure [74].

COPD is characterized by systemic inflammation, airflow limitation, and episodes of acute exacerbation [88]. Severe exacerbations may progress to acute respiratory failure, requiring invasive and noninvasive ventilation. In patients requiring a ventilator, COPD is a risk factor for PMV [89]. Overall, 45–60% of patients with PMV have been reported to have COPD [66,69].

Pulmonary emphysema, diaphragm flattening, and dynamic hyperinflation increase the work of breathing and may result in diaphragm dysfunction in COPD [90,91]. The structure and function of the diaphragm are impaired in COPD, including through sarcomere disruption, reduced strength, and fatigue [92,93]. Systemic factors, such as inflammatory cytokines (e.g., IL-6 and tumor necrosis factor-α), hypoxia, and oxidative stress lead to diaphragm muscle atrophy and an increased proportion of fatigue-resistant type I slow-twitch fibers in COPD [92,93]. Oxidative stress activates proteinase in the diaphragm, reducing diaphragm fiber sensitivity to calcium ion, thus resulting in increased protein degradation and diaphragm atrophy [92,94]. Mitochondrial damage is associated with diaphragm dysfunction in COPD because the electron leakage of the electron transport chain and the production of ROS increase oxidative stress [95,96,97]. Diaphragm atrophy occurs in mild-to-severe COPD, thus increasing the risk of VIDD during PMV. When the severity of COPD progresses, oxidative stress, and mitochondrial dysfunction increase with an increase in oxidative type I fibers in the diaphragm [92,93].

### 5.3. Potential Therapeutic Strategies

Patients requiring PMV often experience long-term immobilization complications, including limb muscle atrophy, diaphragm dysfunction, pressure ulcers, joint contracture, and deconditioning [98,99]. Muscle-strengthening activities aim to progressively improve mobility and functional activity [100]. Pulmonary rehabilitation has been used to improve physical capacity and the quality of life in patients with chronic pulmonary diseases [101,102,103,104]. For patients requiring PMV, pulmonary rehabilitation could provide clinical benefits [100,105,106,107,108]. Management of PMV is summarized in Table 2.

### 5.4. Physical Exercise Programs

A comprehensive physical exercise program includes cardiopulmonary endurance exercises and peripheral muscle training. Upper and lower limb exercises include passive leg raising, weighted resistance, and stationary cycle ergometry training [100,109,110]. Respiratory muscle training consists of placing a sandbag on the abdomen during breathing, using a threshold device, or performing diaphragmatic breathing control [100,108,109,110,111,112,113]. The goal is for the patient to increase their mobility and functional activity and gradually recover from being bedridden to sitting, standing, and walking [106,109,112,113]. The benefits of physical exercise programs for patients requiring PMV include functional improvement, increased weaning rates, shorter duration of hospitalization, and reduced mortality [106,109,112,113]. Notably, Dong et al. showed that early rehabilitation training lessened the diaphragm dysfunction during ventilator use, increased weaning rates from the ventilator, and shortened the intubation duration in patients with MV in the ICU [109]. Additionally, inspiratory muscle training through progressive threshold loadings, such as flow-dependent inspiratory resistive training or pressure-threshold inspiratory muscle strength training can be implemented to increase both inspiratory and expiratory muscle strength, exercise capacity, and successfully wean patients from MV [110,111,112].

### 5.5. Physiotherapy with Positive Pressure

In patients requiring PMV, immobility and long-term ventilator use result in complications, such as the atrophy of respiratory muscles, decreased lung volume, and atelectasis [8]. When performing rehabilitation exercises, patients requiring PMV may experience respiratory muscle fatigue and intolerance to exercise because of the increased ventilatory demand [105]. Additional pressure support (PS) levels can increase tidal volume, reduce the work of breathing, prevent early airway collapse during expiration, and improve exercise performance [105]. In a study, adding 4 cmH_2_O to the baseline PS level increased the exercise duration in patients requiring PMV, especially in those with low respiratory muscle power [105]. Inspiratory muscle weakness is commonly observed in these patients [114,115], which may be improved by adding PS levels during exercise.

Intermittent positive pressure breathing (IPPB) can increase the delivery of inspiratory positive pressure to the airway to achieve homogeneous gas distribution, recruit collapsed alveoli, and facilitate lung expansion [116]. During the weaning process, IPPB and positive end-expiratory pressure (PEEP) may increase lung volume and reduce the work of breathing during the expiratory phase, thus helping patients requiring PMV to tolerate the rehabilitation program [117]. A study demonstrated that a combination of IPPB and PEEP therapy increased strength in the inspiratory muscles and led to a shorter duration of ventilator use [117]. For patients requiring PMV, IPPB could be used as a supplementary therapy to prevent atelectasis and improve weaning outcomes [117].

### 5.6. Electrical Muscle Stimulation Therapy

Electrical muscle stimulation triggers muscle contraction and could be used to recover muscle atrophy and improve muscle strength after prolonged bed rest [118,119,120,121].

The majority of patients requiring PMV have increased muscle catabolism and decreased muscle mass synthesis after prolonged bed rest, particularly in the lower limbs [122,123]. In a study, a daily electrical muscle stimulation program (30 min/session for 10 days) improved muscle strength and conserved leg circumference in patients requiring PMV, but no significant improvement was observed in the weaning or mortality rate [124]. Electrical muscle stimulation can be used to prevent muscle weakness in these patients. A study of critically ill patients on MV demonstrates that induced contraction of the diaphragm by pacing the phrenic nerve not only reduces the rate of its atrophy during MV but also leads to an increase in its thickness [124]. Recent studies have also demonstrated that the implementation of transvenous diaphragm pacing of the phrenic nerve may facilitate liberating patients with VIDD [124].

### 5.7. Nutrition and Antioxidants

Malnutrition is common in PMV patients and associated with poor outcomes, including difficult wound healing, nosocomial infections, and increased mortality [62]. Nutritional support is important, however, there is no standardized protocol regarding the administration route, type of nutrients and timing of nutrition support.

Although studies suggest that the use of N-acetylcysteine and specific mitochondrial-targeted peptides (i.e., SS-31) can protect against muscle atrophy during long-term inactivity, additional studies are needed to determine whether mitochondrial-targeted treatments are effective in specific cellular locations and transporting across multiple biological barriers to prevent muscle weakness during a long duration of muscle inactivity [19,31,40].

## 6. Future Research

Because there’s a lack of wide scientific investigation in the field of PMV and diaphragm dysfunction, future research should involve a shared definition of this condition for the appropriate recruitment of patients. Large, multicentric, multinational RCT are also needed on pathophysiology, molecular pathways (VIDD, exerkine apelin), drug therapy, diaphragmatic pacing, immune modulation and rehabilitative programs for these patients, and need to be distinct from acute critical illness patients.

## 7. Conclusions

The number of patients requiring PMV globally is increasing, with consequences for healthcare systems. Numerous studies have proposed risk and prognostic factors for PMV, and the underlying pathogenic mechanisms include an abnormal respiratory drive and VIDD. After PMV, the diaphragm and limb muscles may exhibit muscle atrophy and impaired contractile force, resulting from accelerated protein degradation. Mitochondrial dysfunction and increased oxidative stress have been identified in animal studies mimicking sepsis and multisystem organ failure. Because multiple comorbidities are associated with PMV, physicians should treat the underlying diseases, prescribe early mobilization and avoid the risk of complications. Furthermore, adequate preventive and therapeutic strategies should be used early to attenuate the effects of PMV on the diaphragm and limb muscles. Research on the mechanisms, prognosis, and treatment (pharmacological agents and rehabilitative programs) of PMV is required for improved management.

## Figures and Tables

**Figure 1 jcm-11-02451-f001:**
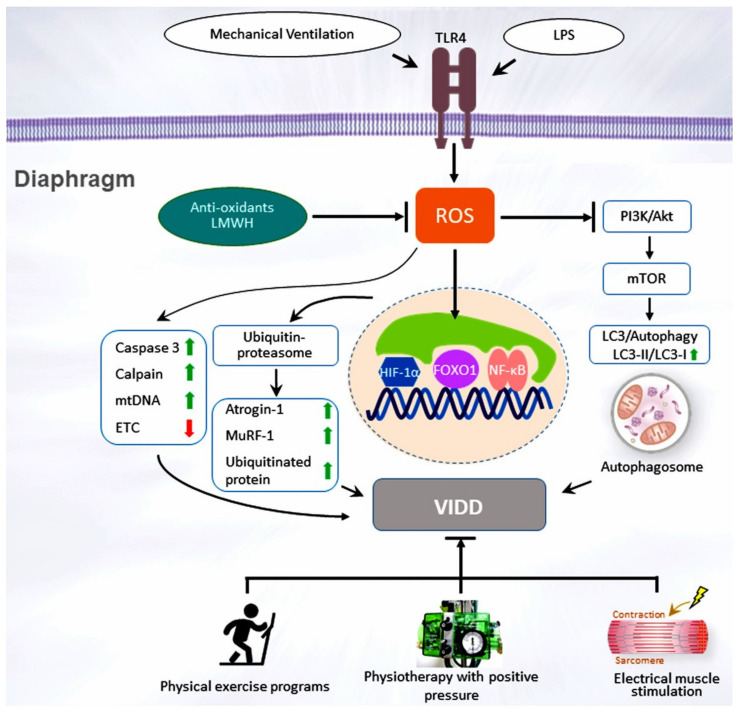
Schematic figure illustrating the signaling pathway implicated in VIDD development. Endotoxin-induced augmentation of mechanical stretch-mediated ROS generation and diaphragm injury are associated with diaphragm proteolysis, mitochondrial dysfunction, autophagy, as well as activation of the caspase-3, calpain, and ubiquitin–proteasome pathways. Diaphragm weakness can be attenuated by administering antioxidants, enoxaparin, or through partial support for mechanical ventilation or pulmonary rehabilitation. Akt = serine/threonine kinase/protein kinase B; ETC = electron transport chain; FoxO1 = Class O of forkhead box1; HIF = hypoxia-inducible factor; LC3 = light chain 3; LMWH = low-molecular-weight heparin; LPS = lipopolysaccharide; mtDNA = mitochondrial DNA; mTOR = mammalian target of rapamycin; MuRF-1 = muscle ring finger-1; NF-κB = nuclear factor κappa B; PI3-K = phosphoinositide 3-OH kinase; ROS = reactive oxygen species; TLR4 = toll-like receptor 4; VIDD = ventilator-induced diaphragm dysfunction.

**Table 1 jcm-11-02451-t001:** Risk factors of prolonged mechanical ventilation.

Systemic comorbidities
Chronic respiratory diseases: COPD, bronchiectasis, pulmonary fibrosis
Heart failure
Cerebrovascular diseases
Neuromuscular diseases
End-stage renal disease
Liver cirrhosis
Malignancy
Infection: sepsis, multi-drug resistant infection
Malnutrition
Ventilator-induced diaphragm dysfunction
Critical illness neuromyopathy
Critical illness encephalopathy

Abbreviation: COPD: chronic obstructive pulmonary disease.

**Table 2 jcm-11-02451-t002:** Management of prolonged mechanical ventilation.

Systemic comorbidities treatment
Infection treatment
Nutrition support
Physical exercise programs
Breathing control
Passive leg raising
Weighted resistance
Stationary cycle ergometry training
Respiratory muscle training
Active limb exercise
Physiotherapy with positive pressure
Additional pressure support during exercise
Intermittent positive pressure breathing during exercise
Cough augmentation techniques
Electrical muscle stimulation therapy

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
