# Peer review of "Prolonged Mechanical Ventilation: Outcomes and Management"

_jcm, 2022, doi:10.3390/jcm11092451_

Round 1

Reviewer 1 Report

Very comprehensive review of prolonged mechanical ventilation. Overall covers various aspects of PMV and well written.  

Just a few comments

Can the authors expand the description of the normal diaphragm, diaphragmatic function, and how trans-diaphragmatic pressure determines function?

In the clinical section can the can the authors describe how to evaluate the respiratory drive bedside- airway occlusion pressure and hypercapnic respiratory drive.

Can the authors describe the various methods to assess diaphragmatic function – measuring pdi, the potential role of neurally adjusted ventilatory assist (NAVA) ventilator mode and EMG.

Any images showing the cellular mechanisms of diaphragmatic dysfunction would add more to the manuscript.

A table with various etiologies/ categories of PMV would be nice

Any diagnostic algorithm for the clinicians to identify at risk patients ?

In the future therapy section- any data on exerkine apelin pathways ?

Adding images/figures/tables would add more overall value. 

Author Response

Very comprehensive review of prolonged mechanical ventilation. Overall covers various aspects of PMV and well written. 

Just a few comments

Can the authors expand the description of the normal diaphragm, diaphragmatic function, and how trans-diaphragmatic pressure determines function?

Response:

Thanks for the reviewer’s comment. We add the description of the normal diaphragm, diaphragmatic function as follows.

The diaphragm is a thin, dome-shaped muscular and central membranous tendon structure that separates the thoracic and abdominal cavities. Under normal conditions of the diaphragm increases the internal height of the and causes of air. Relaxation of the diaphragm and the thoracic cage leads to. The diaphragm strength is measured by transdiaphragmatic pressure (Pdi) derived from the difference between the pressure in the stomach (gastric pressure, Pga) and the esophageal pressure (Pes, as an indicator of pleural pressure): Pdi = Pga – Pes. The gold standard in patients who cannot cooperate in the test is to calculate Pdi via by twitch magnetic stimulation of the phrenic nerves (Pdi,tw). Maximal inspiratory pressure (MIP) was recorded following a maximum inspiratory effort against a occluded airway.21,22,24  

In the clinical section can the can the authors describe how to evaluate the respiratory drive bedside- airway occlusion pressure and hypercapnic respiratory drive.

Response:

Thanks for the reviewer’s comment. We add the description of airway occlusion pressure and hypercapnic respiratory drive.

P0.1 is defined as the negative airway pressure generated during the first 100 ms of an occluded inspiration measured by ventilator

Hypercapnic challenge test is performed with the Modified Read rebreathing method and hypercapnic challenge with CO2 induces an increase in ventilatory drive and the level of P0.1 recorded by bedside capnograph and pneumotachograph.

Can the authors describe the various methods to assess diaphragmatic function – measuring pdi, the potential role of neurally adjusted ventilatory assist (NAVA) ventilator mode and EMG.

Response:

Thanks for the reviewer’s comment. We add the description of pdi, NAVA and EMG.

The diaphragm strength is measured by transdiaphragmatic pressure (Pdi) derived from the difference between the pressure in the stomach (gastric pressure, Pga) and the esophageal pressure (Pes, as an indicator of pleural pressure): Pdi = Pga – Pes. The gold standard in patients who cannot cooperate in the test is to calculate Pdi via by twitch magnetic stimulation of the phrenic nerves (Pdi,tw). Maximal inspiratory pressure (MIP) was recorded following a maximum inspiratory effort against a occluded airway.21,22,24

Furthermore, the introduction of neurally adjusted ventilatory assist (NAVA), a ventilator mode that synchronizes MV to diaphragm electrical signal using an esophageal catheter with multiple electrodes, has facilitated the application of electromyography in the continuous monitoring of diaphragm activity.24 

Any images showing the cellular mechanisms of diaphragmatic dysfunction would add more to the manuscript.

Response:

Thanks for the reviewer’s comment. We add a figure showing the cellular mechanisms of diaphragmatic dysfunction.

A table with various etiologies/ categories of PMV would be nice

Response:

Thanks for the reviewer’s comment. We add a table with various etiologies/ categories of PMV.

Any diagnostic algorithm for the clinicians to identify at risk patients ?

Response:

Thanks for the reviewer’s comment. We add a table with various etiologies/ categories and a table of management for the clinicians to identify at risk PMV patients.

In the future therapy section- any data on exerkine apelin pathways ?

Response:

Thanks for the reviewer’s comment.

Exerkines is humoral factors (peptides, metabolites and RNAs) secreted into circulation after acute exercise or exercise training. Currently, there is no relevant human data in PMV patients and is a field to be investigated. We add the description of exerkines in the future therapy section.

Adding images/figures/tables would add more overall value.

Response:

Thanks for the reviewer’s comment. We add figure and tables in the manuscript.

Reviewer 2 Report

Huang et al. have written a nice review entitled "Prolonged mechanical ventilation: outcomes and pathophysiology." It was a pleasure to read this. This review is systematically organized and includes elements of basic and clinical research. I think it fits well into the publications of the Journal of Clinical Medicine.

Last but not least, a few comments:

- Lines 39-40 describe the incidence of patients in Taiwan who received prolonged mechanical ventilation. Is there more current data on this in 2022?

The authors are from Taiwan.

- Line 48. In my opinion, thought should be given to calling this section "respiratory drive". Respiratory rate has a significant impact on respiratory drive. I think the authors should also explain "mechanical power" and "work of breathing" in this section.

- Line 95-151. To better understand the negative and inflammatory factors influencing the diaphragm during invasive ventilation, an overview chart would be helpful.

- Line 161. How was enoxaparin administered to how many patients in this study?

- Line 156. Please complete the formula for the rapid shallow breathing index. Perhaps you should briefly mention it in the introduction.

- Potential therapeutic strategies. This section should also address the importance of nutrition for patients receiving invasive or noninvasive ventilation.

- Conclusions. Here, early mobilization of intensive care patients with invasive ventilation should be pointed out. This has been previously described by the authors themselves.

Author Response

Huang et al. have written a nice review entitled "Prolonged mechanical ventilation: outcomes and pathophysiology." It was a pleasure to read this. This review is systematically organized and includes elements of basic and clinical research. I think it fits well into the publications of the Journal of Clinical Medicine.

Last but not least, a few comments:

 Using technology to assess nutritional status and optimize nutrition therapy in critically ill patients

- Lines 39-40 describe the incidence of patients in Taiwan who received prolonged mechanical ventilation. Is there more current data on this in 2022? The authors are from Taiwan.

Response:

Thanks for the reviewer’s comment. We add the current numbers of PMV patients in Taiwan.

In Taiwan, the incidence rate of PMV has increased, from 50,481 patients receiving PMV between 1997 and 2007 to 94,324 patients between 2015 and 2019 in total.

- Line 48. In my opinion, thought should be given to calling this section "respiratory drive". Respiratory rate has a significant impact on respiratory drive. I think the authors should also explain "mechanical power" and "work of breathing" in this section.

Response:

Thanks for the reviewer’s comment. We add the description of mechanical power in respiratory drive section.

Mechanical power is the energy transferred to the respiratory system per unit time, which is the product of respiratory rate, tidal volume, airway pressure pressure-controlled ventilation. Higher mechanical power is associated with ventilator induced lung injury and worse clinical outcomes. Mechanical power is a novel concept to guide mechanical ventilator adjustments of ARDS. For patients with PMV, mechanical power may be a tool as respiratory drive and need further investigation.

- Line 95-151. To better understand the negative and inflammatory factors influencing the diaphragm during invasive ventilation, an overview chart would be helpful.

Response:

Thanks for the reviewer’s comment. We add a figure showing the cellular mechanisms of diaphragmatic dysfunction.

- Line 161. How was enoxaparin administered to how many patients in this study?

Response:

Thanks for the reviewer’s comment.

A randomized clinical trial including 20 patients demonstrated that the administration of subcutaneous enoxaparin improv serial gas exchange, reduced D-dimer levels, and resulted in a higher per-centage of successful weaning in patients with severe COVID-19 requiring PMV.

- Line 156. Please complete the formula for the rapid shallow breathing index. Perhaps you should briefly mention it in the introduction.

Response:

Thanks for the reviewer’s comment. Rapid shallow breathing index is calculated as the ratio of tidal volume in liters to respiratory rate in breaths/minute. We add the formula in the introduction.

- Potential therapeutic strategies. This section should also address the importance of nutrition for patients receiving invasive or noninvasive ventilation.

Response:

Thanks for the reviewer’s comment. We add the description of nutrition in potential therapeutic strategies.

Malnutrition is common in PMV patients and associated with poor outcomes, including impaired wound healing, higher rates of nosocomial infections, and all-cause mortality. Nutritional support is important but there is no standardized protocol about administration route, type of nutrients and timing of nutrition support.

- Conclusions. Here, early mobilization of intensive care patients with invasive ventilation should be pointed out. This has been previously described by the authors themselves.

Response:

Thanks for the reviewer’s comment. We add early mobilization of intensive care patients with invasive ventilation in conclusions.